# Ethacrynic Acid: A Promising Candidate for Drug Repurposing as an Anticancer Agent

**DOI:** 10.3390/ijms24076712

**Published:** 2023-04-04

**Authors:** Lu Yu, Ho Lee, Seung Bae Rho, Mi Kyung Park, Chang Hoon Lee

**Affiliations:** 1College of Pharmacy, Dongguk University, Seoul 04620, Republic of Korea; 2Department of Cancer Biomedical Science, Graduate School of Cancer Science and Policy National Cancer Center, Goyang 10408, Republic of Korea

**Keywords:** ethacrynic acid, cancer hallmarks, lung cancer, ion channel, glutathione, Wnt signaling

## Abstract

Ethacrynic acid (ECA) is a diuretic that inhibits Na-K-2Cl cotransporter (NKCC2) present in the thick ascending loop of Henle and muculo dens and is clinically used for the treatment of edema caused by excessive body fluid. However, its clinical use is limited due to its low bioavailability and side effects, such as liver damage and hearing loss at high doses. Despite this, ECA has recently emerged as a potential anticancer agent through the approach of drug repositioning, with a novel mechanism of action. ECA has been shown to regulate cancer hallmark processes such as proliferation, apoptosis, migration and invasion, angiogenesis, inflammation, energy metabolism, and the increase of inhibitory growth factors through various mechanisms. Additionally, ECA has been used as a scaffold for synthesizing a new material, and various derivatives have been synthesized. This review explores the potential of ECA and its derivatives as anticancer agents, both alone and in combination with adjuvants, by examining their effects on ten hallmarks of cancer and neuronal contribution to cancer. Furthermore, we investigated the trend of synthesis research of a series of ECA derivatives to improve the bioavailability of ECA. This review highlights the importance of ECA research and its potential to provide a cost-effective alternative to new drug discovery and development for cancer treatment.

## 1. Introduction

Cancer has emerged as a significant public health concern, posing a grave threat to people’s well-being worldwide. The latest data estimate that in the United States alone, 1,958,310 new cancer cases and 609,820 cancer-related deaths will occur by 2023 [1]. The figures for 2023 indicate a further 40,280 new cases compared to 2022, highlighting the urgent need for effective treatment options [1,2].

Several factors, including an aging population, unhealthy lifestyle choices, inadequate medical facilities, poverty, and lack of cancer screening, have contributed to the proliferation of malignant tumors [3]. Promoting healthy lifestyle habits and avoiding harmful practices could help prevent and control more than half of all cancers [4].

In recent years, innovative anticancer drugs, particularly immuno-oncology drugs such as checkpoint inhibitors, have shown promise. However, there are still several limitations that necessitate combining these drugs with other anticancer agents, such as immune checkpoint blockers or drugs in molecular targeted therapy [5]. The exorbitant costs of developing antitumor drugs have also imposed a significant financial burden on many countries and patients [6,7,8].

Drug repositioning, or repurposing existing drugs for cancer treatment, offers a promising risk-reward trade-off. This strategy represents a cost-effective alternative to new drug discovery and development as it can help reduce development costs, shorten development time, and mitigate clinical risks [9]. Ethacrynic acid (ECA), commonly used as a diuretic in clinical practice, has been touted as a promising candidate for drug repositioning as an anticancer drug [10,11,12].

## 2. What Is the Ethacrynic Acid?

Ethacrynic acid (ECA), also known as Edecrin or EA, is a cyclic diuretic that was discovered over 50 years ago and is used in clinical practice to promote the excretion of excess water and salts from the body in urine [13,14,15,16]. ECA acts as an SH reagent, reacting with -SH residues of GSH and other proteins such as cysteine, imidazole, histidine, and lysine [17]. It has a molecular formula of C_13_H_12_C_l2_O_4_, a molecular weight of 303.14 Da, and appears as a white or almost white crystalline powder (Figure 1).

ECA is available as 25 mg tablets and solutions and is approved by the US FDA for the treatment of edema, congestive heart failure, and high blood pressure. It is often administered orally or intravenously due to its low bioavailability [15]. The recommended dose for adults is 25–100 mg orally one to three times daily. After intravenous administration, ECA has a mean elimination half-life of 30 min (12–160 min) and increases the transport of Na^+^ into the distal tube, leading to a rapid and profound diuretic effect that excretes large amounts of sodium and chloride [15,18]. 

ECA is a widely used diuretic that directly causes smooth muscle vasodilation [15,16,19,20]. However, its clinical use is limited by known side effects and toxicities, including ototoxicity, which can disrupt the blood–cochlear barrier, and cholestatic jaundice [13,20,21,22,23]. Co-administration of ECA with cisplatin or gentamicin should be avoided, as it initiates the caspase-9 pathway and rapidly destroys cochlear cells, significantly enhancing the ototoxic effects of cisplatin and/or gentamicin [24,25]. Furthermore, prolonged use of ECA may cause gastrointestinal symptoms such as abdominal discomfort, vomiting, and profuse watery diarrhea [20].

Despite these limitations, ECA remains a useful diuretic for sulfonamide-sensitive patients, as it is the only circulating diuretic that does not contain a sulfonamide moiety and thus avoids severe allergic reactions [20]. However, its use requires rigorous monitoring, and alternative diuretic options should be considered in patients with known contraindications.

## 3. Effects of ECA on Cancer Hallmarks

Cancer is characterized by ten hallmarks that include maintaining proliferative signaling, resisting cell death, achieving replicative immortality, and activating invasion and metastasis, among others [26]. These hallmarks can be divided into those originating from the cancer cells themselves and those related to the tumor microenvironment. Nerves in cancer cell tissues have also been recognized as one of the tumor microenvironments. Therefore, the effect of ECA on nerves was also considered and discussed. ECA, a potential anticancer agent, has been shown to reverse drug resistance, inhibit cell proliferation, induce cancer cell death, and inhibit migration [27]. In this section, we summarize and analyze the effects of ECA on several cancer hallmarks from cancer cells themselves (Figure 2).

### 3.1. ECA Suppresses Proliferation and Growth of Tumor

Cancer cells are characterized by their ability to proliferate indefinitely, which is a key factor in tumor growth and metastasis [28].

ECA has been found to exhibit antitumor effects in a range of cancer cells, with varying cytotoxicity (IC_50_ values range from 6 to 223 μM, Table 1). Studies have demonstrated that ECA inhibits the survival of cancer cells in a time- and dose-dependent manner and has an antiproliferative effect on tumor cells. Early research showed that ECA reversibly inhibits the proliferation of human leukemia cells by inhibiting GST activity [29]. Subsequent studies have confirmed ECA’s antiproliferative effects on a range of cancer cell lines, including pancreatic, prostate, ovarian, lung, breast, bladder, renal, and bone marrow-derived primary myeloma cells [27,30,31,32,33,34,35,36,37,38].

While low concentrations of ECA promote cell proliferation, high concentrations have been found to have an antiproliferative effect on tumor cells [37,38]. ECA’s cytotoxicity varies among different cancer cells, with breast cancer, malignant glioma, and prostate cancer cells being more sensitive [34]. 

ECA has been demonstrated to possess a significant in vivo inhibitory effect on tumor growth in several studies [12,27,32,36,38,45]. For instance, ECA reduces tumor growth in mouse xenograft models of myeloma cells, prostate carcinoma cells, and orthotopic mouse models of NOD/SCID lung cancer cells [12,45]. In particular, injection of 50 mg/kg of ECA results in a 49% reduction in tumor weight of DU145 cells in mice [32]. Furthermore, injection of 3 mg/kg ECA-treated A549 cells in mice results in a 50% reduction in the surface area of lung cancer compared to control mice [12]. In another study, orally treating mice with 450 μg/day of ECA for 60 days significantly reduces tumor growth and increases overall survival [45]. 

Combination therapy, which involves the use of two or more drugs, is often employed to achieve a more effective therapeutic outcome than with a single drug. In the case of breast cancer and non-small cell lung cancer, the combination of ECA and afatinib has been shown to have a potent inhibitory effect on cell proliferation [30,38]. Notably, the inhibitory effect of ECA and afatinib on lung cancer cells is higher than that of either drug alone [38]. Similarly, ECA and neratinib (4 µM) in combination have been found to have an antiproliferative effect on breast cancer cells, although the effect is not as pronounced as with ECA and afatinib [30]. ECA has also been found to enhance the cytotoxicity of certain compounds, such as trans-cinnamaldehyde and methyl cinnamyl ketone, leading to a significant reduction in the viability of human breast cancer cells [46].

In mouse models of myeloma and lung cancer, ECA, in combination with other drugs, such as lenalidomide and afatinib, has demonstrated increased antiproliferative activity compared to single-drug treatments [36]. Furthermore, ECA has been found to effectively inhibit tumor growth when administered in a nanoparticle-mediated manner to sensitize Pt drugs, as well as to reverse resistance to anticancer drugs [47].

Overall, these findings suggest that combination therapy with ECA and other drugs may offer a promising approach to inhibiting cell proliferation and tumor growth in various types of cancer.

### 3.2. ECA Induces Cell Death

Resistance to cell death is a hallmark of cancer, and the ability to induce cell death is a key feature of successful cancer therapies [28]. Apoptosis, or programmed cell death, is a normal process that helps maintain healthy tissues by eliminating damaged or infected cells. Cancer cells, however, are able to evade apoptosis and continue to proliferate abnormally [28]. Thus, the development of cancer therapies that induce cancer cell apoptosis represents a promising avenue for cancer treatment.

Recent research has shown that ECA, a drug, selectively induces apoptosis in a variety of cancer cell lines, including breast, colon, liver, lung, melanoma, and lymphoma cells [30,35,36,37,40,42,43,48,49].

ECA’s ability to induce cell death varies among different cancer cells, with varying concentrations needed for maximum effect. For example, ECA induced cell death in human colon cancer DLD-1 cells at high concentrations of 60–100 µM [37], while it had the greatest effect on the apoptosis of colon cancer cells at a concentration of 200 µM [27]. Schmidt et al. found that an ECA treatment concentration of 30 μM was the most effective concentration for lymphoma and myeloma cell death [44].

Studies have also demonstrated that ECA can synergize with other chemotherapeutic agents to induce cancer cell apoptosis. For example, when combined with afatinib, ciclopirox olamine (CIC), piceatannol (PIC), or arsenic trioxide (ATO), ECA effectively induces cancer cell death [27,38,42,43,44,50]. In addition, ECA was found to inhibit cell cycle progression and induce apoptosis in EGFR L858R/t790m-mutated lung cancer cells at a concentration of 75 μM [38].

While ECA has been shown to induce cell death in cancer cells, it has also been found to promote apoptosis in normal cells. For instance, ECA induced PC12 neuronal cell death [51] and promoted NSC-34 motor neuron apoptosis by inducing the expression of oxidative stress marker HO-1 [52]. Nevertheless, the mechanism by which ECA inhibits tumor cell growth and induces tumor cell apoptosis at high concentrations remains unclear, as noted by some researchers [37,51].

Combination therapies that include ECA have also shown promise in enhancing cancer cell death. ECA, in combination with afatinib, significantly reduced the G0/G1 phase, blocked the G2/M phase, and increased the apoptosis rate of lung cancer cells by two-fold [38]. ECA also enhances the antitumor effect of afatinib. Additionally, the combination of ECA with other drugs such as ciclopirox olamine, piceatannol, arsenic trioxide, lenalidomide, or thalidomide has resulted in increased cytotoxicity against different cancer cell lines, including liver, pancreatic, prostate, ovarian, bladder, bronchial, renal, and colon fibroblast cells [27,36,42,43,44]. 

Although the exact mechanism by which ECA induces tumor cell apoptosis is unclear, its ability to selectively induce cell death in a variety of cancer cells makes it a promising candidate for cancer therapy. Further research is needed to elucidate its mechanisms and potential clinical applications.

### 3.3. Effects of ECA on Replicative Immortality

Replicative immortality, a hallmark of advanced malignancies, is associated with the reactivation of telomerase and provides the basis for the proliferation of cancer cells [28]. Telomerase is activated in cancer cell immortalization, providing unlimited replication potential to cancer cells [53]. While the effect of ECA on telomeres or telomerase is currently unknown, studies suggest that Myc activates telomerase and regulates its activity [54]. Upregulation of c-Myc induces enhanced telomerase activity [55], and MYC directly activates the transcription of hTERT, the catalytic subunit of telomerase [55,56]. Interestingly, ECA has been reported to downregulate the expression of MYC [38], suggesting that it may regulate telomerase expression through the downregulation of c-MYC. Additionally, evidence suggests that Ca^2+^ homeostasis is associated with telomerase activity [57], and ECA has been shown to inhibit L-type voltage-dependent calcium channels [20]. Thus, ECA may be involved in the replicative immortality of cancer cells by regulating telomerase activity. However, further studies are needed to fully understand the effects of ECA on telomeres and telomerase.

### 3.4. ECA Increases the Efficiency of Tumor Suppressors

The evasion of tumor suppressors is a key characteristic of cancer, which involves the inactivation of tumor suppressors that regulate cell division and replication. Among these growth suppressors, p53 is a crucial transcription factor that inhibits cell growth and proliferation, and its targeted therapy represents a promising approach to cancer treatment [58,59]. In this context, ECA has been identified as a p53 activator in human cancer cells, which inhibits GSTP1 and leads to an increase in basal p53 levels [60,61]. This effect has important implications for improving the efficacy of anticancer drugs by restoring p53 responsiveness through GSTP1 inhibition [61,62]. Indeed, immunofluorescence microscopy has shown that ECA-treated cells have a higher nuclear expression of p53 [62]. Thus, ECA can activate the growth inhibitor p53 and inhibit tumor growth, indicating its potential as a therapeutic agent for cancer treatment.

### 3.5. Effects of ECA on Genome Instability and Mutation

Genomic instability is a hallmark of cancer, with mutations playing a major role in tumorigenesis [28]. Abnormal proliferation of cancer cells leads to increased genomic mutations, causing the loss of control of key cellular functions and resulting in genomic instability [28]. While the direct link between ECA and tumor genomic instability is unclear, research has shown that ECA targets functional Cys257 in adenine nucleotide translocases (ANTs), inducing impaired mitochondrial function and promoting the death of A375 human malignant melanoma cells and HeLa cervical cancer cells [40]. 

Mutations in p53 are the most common genetic alterations in human cancers, and tumors with p53 mutations often have a poor prognosis [59,63]. However, small molecule derivatives of ECA, such as KSS72, have been shown to reactivate the mutant p53 protein, restoring wild-type form structure and function by covalently binding p53R248 protein, thereby inhibiting tumor growth [64]. We speculate that ECA may have a similar effect on gene mutation as its derivatives, such as KSS72.

### 3.6. ECA and Deregulating Cellular Energetics

Reprogramming of energy metabolism is a hallmark of cancer that promotes cell growth and differentiation, ultimately contributing to the rapid growth, division, and proliferation of cancer cells [28]. This altered metabolism requires more energy supply than normal cells to maintain redox homeostasis. The primary metabolic pathway used by cancer cells for energy production is glycolysis, also known as the Warburg effect. However, some cancer cells rely on mitochondrial oxidative phosphorylation as the main source of ATP production [65].

Interestingly, even under aerobic conditions, cancer cells metabolize glucose to pyruvate and then undergo aerobic glycolysis to lactate instead of the more efficient oxidative phosphorylation pathway. This suggests that metabolic disorders are associated with cancer and that metabolic reprogramming and tumor development are mutually causal. Activation of oncogenes or inactivation of tumor suppressor genes is known to reprogram the metabolism of cancer cells [66], enhancing cancer progression not only via pathways promoting growth but also by avoiding cell death [67]. This view is also supported by Gordon and de Hartog et al. [68].

One known ATPase inhibitor that disrupts cellular processes such as mitochondrial metabolism and glycolysis is ECA. Studies have shown that ECA moderately impairs energy production, inhibits glycolysis, and significantly blocks lactate formation [68,69]. ECA also interferes with mitochondrial oxidative phosphorylation and membrane ATPase, thereby limiting energy for sodium transport. ECA inhibits sodium- and potassium-dependent ATPase by affecting two distinct steps of enzymatic turnover. This inhibition of Na^+^, K^+^ activation, and Mg^2+^-dependent ATPase activity by ECA is irreversible.

Mitochondrial bioenergetics is required for tumorigenesis [62], and ECA induces mitochondrial shape change and redistribution, preserves mitochondrial transmembrane potential, and stimulates mitochondrial respiration [70]. However, ECA also hinders ATP production and impairs mitochondrial function [71], which may be related to the depletion of intracellular mitochondrial GSH. ECA directly depletes intracellular GSH to generate concentration-dependent reactive oxygen species intermediates that affect mitochondrial function [51]. Additionally, in astrocytes, ECA inhibits GSH and induces a significant increase in intracellular ROS [72]. Loss of mitochondrial GSH may lead to disturbances in mitochondrial function and energy metabolism [72].

ECA interferes with glycolysis and mitochondrial metabolism, alters the morphology and function of mitochondria, and reduces ATP production, thus affecting the energy metabolism of cancer cells. These findings suggest that ECA may have potential in the development of cancer therapies that target metabolic reprogramming.

### 3.7. ECA Impedes Metastasis and Invasion

Invasion and metastasis are complex biological processes that are critical for the progression of cancer [28]. These processes involve the ability of tumor cells to invade surrounding tissues and enter into the blood and lymphatic vessels, ultimately leading to the colonization of distant organ sites [28,73]. The loss of cell adhesion and changes in biomechanical and biophysical properties of cells are key drivers of cancer cell metastasis. Metastasis is a major cause of mortality in cancer patients, making it imperative to identify novel antimetastatic compounds [74].

Ethacrynic acid (ECA) has recently been shown to impede tumor invasion and metastasis by inhibiting the perinuclear reorganization of keratin 8 filaments and decreasing the viscoelasticity of cancer cells [75]. This leads to a decrease in cancer cell migration and invasion. ECA also inhibits epithelial–mesenchymal transition, a process that is associated with the loss of the cell adhesion molecule E-cadherin and an increase in N-cadherin, resulting in increased cell migration. ECA restores the expression of E-cadherin and N-cadherin, further decreasing cancer cell migration and invasion. ECA inhibited metastasis of lung cancer cells via the blocking of NDP-induced Wnt signaling [12]. Interestingly, ECA inhibited EMT at a lower concentration than that inhibited the growth of cancer cells [12,30,37,42,44]. 

ECA has been found to inhibit GSTO1 activity, a protein that is involved in cancer cell migration. GSTO1 inhibition has been shown to attenuate migration by inhibiting the activation of the JAK/STAT3 signaling pathway [76]. Clinical studies have reported that ECA has weakly inhibited migration and has been repurposed for the treatment of nonmuscle invasive bladder cancer [35]. Additionally, new ECA derivatives have been synthesized, which significantly inhibit migration in breast and prostate cancers [77,78]. 

Taken together, these findings suggest that ECA could be used as a promising antimetastatic drug. Further research is needed to fully elucidate the molecular mechanisms underlying the inhibitory effects of ECA on tumor invasion and metastasis.

### 3.8. ECA Hinders Angiogenesis

Angiogenesis is a critical step in tumor progression, which involves the development of new blood vessels from pre-existing ones [79,80]. The growth and spread of tumors rely on angiogenesis to obtain nutrients and oxygen and to expel waste products. Consequently, inhibition of tumor angiogenesis has emerged as a promising therapeutic approach to prevent tumor progression. Angiogenesis is regulated by both activator and inhibitor molecules [79]. The combination of antiangiogenic drugs with immunotherapy represents a potential therapeutic strategy [80].

Hypoxia plays a critical role in inducing cancer angiogenesis, and its key regulator is the hypoxia-inducible factor HIF-1, which promotes angiogenesis by activating target genes that regulate the mitotic and migratory activity of endothelial cells [81,82]. HIF-1 also regulates multiple aspects of tumorigenesis and plays a key role in regulating angiogenic growth factors [82].

ECA has been found to inhibit the HIF pathway by downregulating the mRNA expression of VEGF, a downstream target gene of HIF-1a, in HeLa cells under hypoxia, thereby inhibiting tumor angiogenesis and metastasis [83]. Additionally, ECA has been reported to decrease the expression of the angiogenic factor nerve growth factor (NGF) following prolonged exposure [84]. NGF has been shown to promote angiogenesis by increasing VEGF secretion in cancer cells and targeting NGF through anti-NGF antibodies or small interfering RNA therapies that inhibit tumor angiogenesis [84,85].

These findings suggest that ECA has an inhibitory effect on angiogenesis and may have potential as an anticancer therapy by preventing the growth and spread of tumors through the inhibition of angiogenesis.

### 3.9. ECA and Immune Evasion

Immune evasion is a major hallmark of cancer progression, whereby tumors evade the immune system’s ability to recognize and destroy them [28]. The process of immune evasion occurs in three stages, starting with immunosurveillance and progressing to immune homeostasis and immune escape [86]. Regulatory T cell (T_reg_)-mediated immunosuppression in the tumor microenvironment is a key factor in immune evasion and may pose a barrier to successful tumor immunotherapy [28,87].

Recent studies have identified various molecular and cellular targets related to immune evasion mechanisms, including the promotion and activation of macrophages and the inhibition of Tregs [87]. Tumor-associated macrophages (TAMs) play a crucial role in tumor immune evasion by secreting anti-inflammatory cytokines that contribute to the formation of a tumor-promoting microenvironment, while M1-macrophages promote immune destruction by producing immune killer molecules and inflammatory cytokines [88].

Although no direct studies have investigated the effect of ECA on cancer immune evasion, there is evidence to suggest that ECA may affect macrophages. Research has suggested that ECA inhibits the activated NF-kappaB pathway in murine macrophages [89]. ECA has been shown to improve macrophage function under hyperoxia by attenuating NF-κB activation and restoring hyperoxia-impaired macrophage phagocytic activity and migration [90]. Moreover, ECA has been found to downregulate CYLD in chronic lymphocytic leukemia, and CYLD plays a key role in promoting Treg differentiation and development while maintaining normal T regulatory cell function [91,92]. These findings suggest that ECA may have the potential to reduce tumor evasion from immune destruction by modulating the activity of macrophages and T_reg_ in the tumor microenvironment.

### 3.10. Effects of ECA on Tumor-Associated Inflammation

Inflammation has been identified as a key contributor to the development and progression of various types of cancer. This process occurs through two pathways: the intrinsic pathway, where oncogenic changes create an inflammatory environment that promotes tumor growth [93], and the extrinsic pathway, where chronic inflammation increases the risk of cancer in certain anatomical sites [93,94]. Tumor-associated inflammation plays a crucial role in tumor progression, promoting angiogenesis and metastasis [94].

Anti-inflammatory agents have emerged as a promising approach to cancer therapy. Previous research has indicated that the diuretic agent ethacrynic acid (ECA) possesses anti-inflammatory properties [20]. ECA has been shown to inhibit proliferative inflammation and to induce the expression of phase 2 proteins, which suggests that it inhibits the secretion of the cytokine-like nuclear protein HMGB1 [95].

Multiple studies have demonstrated that ECA inhibits the proinflammatory transcription factor NF-κB pathway, which is a common target of immunosuppressive and anti-inflammatory agents [37,89,95]. This pathway plays a key role in inflammatory diseases, affecting cytokine signaling. ECA has been found to decrease the expression of proinflammatory cytokines in the intestinal wall [96] and to enhance the antitumor effect of afatinib in non-small cell lung cancer by inhibiting the secretion of anti-inflammatory factors [38].

These findings suggest that ECA has the potential as an anti-inflammatory agent that could be useful in diseases associated with excessive inflammation, including cancer. The ability of ECA to inhibit multiple points in the NF-κB pathway and to induce the expression of phase 2 proteins suggests that it could serve as a molecular starting point for the development of new anti-inflammatory agents.

### 3.11. Effects of ECA on Tumor-Related Neural Input

Neuronal inputs play a significant role in the tumor microenvironment, modulating cancer cell behavior and influencing tumor progression [5]. Perineural invasion and infiltration are common features of some tumors, and tumors can reprogram neurons to recruit new nerve fibers [97]. Tumor innervation can release neurotransmitters directly into neural tumor synapses, regulating tumor cell function and affecting cancer progression [98,99]. Ethacrynic acid (ECA) is a compound that has been studied for its effects on neurons, but its direct connection to neuronal input in tumors is yet to be established.

Studies have shown that ECA can induce status epilepticus and enhance excitatory amino acid activity, accelerating the release of noradrenaline and glutamate in the brain [100,101]. ECA also stimulates glutamate release, which can activate glutamate receptors in cancer cells, leading to tumor development [102,103]. Additionally, ECA induces the expression of c-Fos and nerve growth factor NGF, which enhances spontaneous and depolarization-evoked glutamate release from hippocampal nerve terminals [104]. NGF is overexpressed in most human breast and pancreatic cancers, and its overexpression is associated with enhanced proliferation, invasion, and tumorigenicity of cancer cells [105].

ECA’s potential to affect neuronal input in tumors is further supported by its rapid depletion of glutathione, which impairs mitochondrial function and leads to death in motor neurons [52]. c-Fos, a regulator of the epithelial-mesenchymal transition (EMT) process, is also associated with cancer development and metastasis [106,107]. Overexpression of c-Fos enhances EMT status and regulates EMT gene expression, promoting tumor growth [108].

While there is no current evidence showing that ECA acts as a direct neuronal component to promote cancer development, its effects on neurons suggest that it may have an influence on tumor progression. The potential connections between ECA and neuronal input in tumors warrant further investigation to fully understand its role in cancer development and progression.

## 4. Mechanism of Ethacrynic Acid on Cancer

We discuss in this section the mechanism of action of ECA on cancer hallmarks. ECA might suppress the development of cancer through cooperative inhibition through multiple mechanisms (Figure 3).

### 4.1. Blocking of the Ion Channels

Recent studies have shed light on the critical role of ion channels in cancer progression, highlighting the significance of these channels in tumor growth and metastasis [109,110,111]. Ion channels such as calcium, potassium, sodium, and chloride channels have all been found to be involved in cancer cell proliferation, migration, and survival [109,111]. Enhanced expression or kinetics of ion channels in cancer cells are associated with increased malignant potential, further underscoring the importance of these channels in cancer development.

Notably, the voltage-gated L-type calcium channel and the Na^+^/K^+^/2Cl^−^ cotransporter (NKCC) have been identified as highly expressed in colon and gastric cancer, respectively [112,113]. Inhibition of NKCC reduces gastric cancer cell proliferation by affecting the G0/G1 state while blocking voltage ion channels has been found to be an effective approach to prevent tumor growth and metastasis [112,114].

Although direct inhibition of cancer through ion channels by eucalyptol (ECA) has not yet been reported, this compound may hold potential for cancer treatment by inhibiting voltage-gated calcium or sodium channels or blocking NKCC [115,116]. Studies have demonstrated ECA’s ability to inhibit voltage-gated calcium and sodium channels, relax airway smooth muscle cells by inhibiting L-type voltage-dependent calcium channels and store-operated calcium channels, and inhibit the Na^+^/K^+^/2Cl^−^ cotransporter [15].

As such, it is anticipated that future research will uncover the anticancer and cancer metastasis inhibitory effects of ECA through the inhibition of ion channels present in cancer cells. These findings may open up new avenues for the development of novel cancer therapeutics.

### 4.2. Inhibition of the Wnt Signaling Pathway

The Wnt signaling pathway plays a crucial role in various carcinogenic processes, including colorectal cancer, EGFR mutant lung cancer, lymphoma, and multiple myeloma [36,39,117,118,119]. Wnt pathway activation is mediated by the binding of Wnt ligand proteins to the Frizzled and LRP5/6 receptors, leading to the activation of disheveled receptor family proteins and the inhibition of β-catenin degradation by the APC/axin/GSK-3β complex. Stabilized β-catenin is then translocated to the nucleus and transactivates TCF/LEF1 target genes [117,120,121].

Recent studies have shown that ethacrynic acid (ECA) targets β-catenin, which acts as a transcription factor in the Wnt pathway [48]. Specifically, ECA inhibits β-catenin expression by suppressing its promoter activity without binding directly to β-catenin [39,122]. Furthermore, ECA has been demonstrated to suppress the epithelial-mesenchymal transition (EMT) process induced by SPC or TGF-β1 in lung cancer cells by inhibiting Wnt activation via downregulation of NDP expression [12]. NDP is known to act as an atypical Wnt ligand, and previous studies have demonstrated that it binds to Fz4 and LRP5/6, leading to activation of the Wnt/β-catenin pathway [123,124]. Therefore, the observed downregulation of NDP expression by ethacrynic acid may explain its inhibitory effect on the Wnt pathway and downstream β-catenin expression. In a study of EGFR mutant lung cancer, Nakayama and colleagues (2014) found that ethacrynic acid can reduce lung tumor formation in an in vivo mouse model by inhibiting β-catenin in the Wnt signaling pathway [119].

Furthermore, ethacrynic acid was observed to inhibit not only the Wnt3a-mediated Wnt/β-catenin signaling pathway that is secreted as a Wnt ligand but also the Wnt5a/β-catenin, Wnt/LRP6, and Dvl signaling pathways, as reported by Ahmed et al. (2016), Bing Liu et al. (2016), and Lu et al. (2009) [30,39,122]. Interestingly, Wnt5A gene expression has been found to be significantly positively correlated with overall survival in lung cancer patients. Zhang et al. (2021) reported that the combination of ECA and afatinib had a significant impact on the expression of Wnt1/β-catenin and Wnt5A mRNA in H1975 lung cancer cells, although no significant difference was observed in A549 lung cancer cells [38]. 

Abnormal expression and activity of the transcription factor LEF1 have been implicated in tumorigenesis, and LEF1 plays a key role in regulating various cancer-related processes, such as EMT, cancer cell proliferation, migration, invasiveness, and cell death. Moreover, LEF1 has been identified as a valuable biomarker for predicting patient prognosis, with high levels of LEF1 expression indicating a poor outcome [92,125]. Further studies have also demonstrated the ability of ECA treatment to lead to a reduction in LEF1 expression in myeloma MPC-11 cells, as well as a decrease in TCF-4 expression in myeloma OPM-2 cells, as reported by Schmeel et al. (2013) [43]. Interestingly, this reduction in TCF/LEF-β-catenin-specific luciferase reporter activity was also observed in a variety of cancer cell lines, including SW480 and HCT116 colon cancer cells and myeloma cells, in a concentration-dependent manner [36,126,127]. Real-time quantitative PCR experiments have demonstrated that ECA treatment leads to a reduction in the expression of Wnt target genes, including LEF1 and CCND1. In addition, ECA suppresses the expression of LEF1 and CCND1 in cancer cells, according to studies by Ahmed et al. (2016) and Lu et al. (2009) [39,122]. Co-immunoprecipitation experiments have revealed that ECA can directly bind to the LEF-1 protein in SW480 colon cancer cells [39,122] and destabilize the LEF-1-β-catenin complex, thereby interfering with its formation with β-catenin. By inhibiting the transcriptional regulatory activity of the TCF/LEF-β-catenin complex, ECA ultimately suppresses the Wnt pathway [39,122,126,127]. Furthermore, ECA has been shown to reduce the ability of LEF1 to bind to DNA, as observed in studies by Wu et al. (2016) [92]. 

These findings demonstrate the potential of these compounds in the treatment of cancer and offer insights into the complex interplay between the Wnt/β-catenin pathway and tumor development.

### 4.3. ECA as a GSH-Related Enzyme Inhibitor

Increased activity of glutathione transferase (GST) has been observed in numerous human tumors, and its expression is inversely correlated with prognosis, suggesting a potential role in drug resistance [15,128,129,130,131]. According to sequence similarity and substrate specificity, GSTs are classified as alpha (A), kappa (K), mu (M), omega (O), pi (P), sigma (S), theta (T), and zeta (Z) category (Table 2) [132]. ECA mainly acts on alpha (A), omega (O), and pi (P). Overexpression of GST, particularly GSTP1-1, has been linked to innate and acquired resistance to anticancer drugs [133], indicating that cancer cells utilize the GSH-binding activity of GST to evade the effects of these drugs.

Inhibitors of GSTP1-1 have shown promise in counteracting drug resistance [134]. Ethacrynic acid (ECA) is a commonly used inhibitor of GST that also affects glutathione reductase (GR) and regulates detoxification gene expression, leading to increased cellular oxidative stress and GSH levels (Figure 4) [20,29,135,136]. ECA has also been found to affect the half-life of GSTn transcripts and proteins in tumor cells [137]. Despite the promising results of in vitro experiments using ECA to suppress GST regulation and improve response to anticancer drugs, clinical success has been hindered by the side effects of ECA, and further development of GST inhibitors is warranted [138,139].

ECA and its derivatives have been found to inhibit GST activity in various tumors, such as the murine brain tumor, human lung cancer, and human ovarian cancer. In vitro and in vivo studies have demonstrated that ECA can reverse drug resistance phenotypes. These findings have been reported in various studies, including those conducted by Paula B. Caffrey et al. (1999), Peter J. O’Dwyer et al. (1991), Sau et al. (2010), Smith et al. (1989), and T. Rhodes and Twentyman (1992), among others [130,135,140,141,142].

ECA interferes with GST activity either as a direct inhibitor of GST activity or as a substrate for GST-mediated GSH binding, as described by Mark S. Khil et al. (1996) and Wang et al. (2017) [34,143]. Studies have shown that ECA is inactivated by reversibly covalently bound to GSH via GST catalysis using the energy generated from the hydrolysis of ATP, as reported by Ploemen et al. (1994), Wang et al. (2017), and Zaman et al. (1996) [143,144,145]. This binding has been shown to be stereoselective, as described by Marlou L.P.S. van Iersel et al. (1998) [146]. Interestingly, covalent conjugates of ECA and GSH have been demonstrated to be more potent GST inhibitors than ECA as direct inhibitors, as reported in several studies [145,147,148,149].

GSTA1 is a critical factor in cancer progression and is expressed abundantly in most human tumors and tumor cell lines [150,151]. The protein is primarily located in the cytoplasm and cell membrane and is involved in regulating apoptosis signaling. Additionally, GSTA1 has been shown to influence cancer cell viability, EMT processes, adhesion, invasion, and metastasis [152,153].

Cameron and colleagues were the first to elucidate the structure of human GSTA1-1 [150]. Subsequent studies have demonstrated that ECA, as a ligand, can bind to the hGSTA1-1 enzyme through various binding sites and act as a potent inhibitor of hGSTA1-1 [154]. Docking studies have identified multiple binding sites for ECA within the hGSTA11 subunit, including PHE30, VAL28, GLY27, ALA24, SER202, GLY201, LYS196, and PRO200, which differ from those predicted by X-ray data [154].

ECA exhibits nonlinear competitive inhibition of GSH and uncompetitive linear inhibition of CDNB, with uncompetitive inhibitors requiring increased substrate concentrations for inhibition to occur [134]. In the ECA-GSH structure, the electrophilicity of the β-alkene carbon of ECA is enhanced through hydrogen bond formation between ECA ketone oxygen and water molecules with Tyr108 and Asn204, respectively. The hydroxyl group of Tyr108 then adds to Michael ethacrynic acid with glutathione, leading to a catalytic mechanism [149,155]. Kumar and colleagues have suggested that ECA is also involved in hydrogen bonding with residues Gly14 and Met208 and that residue Arg13 forms a salt bridge [155].

In their 2013 study, Musdal et al. found that ECA is a potent inhibitor of GSTA1-1 and GSTA3-3 [134]. ECA’s inhibitory effects on GSTP1-1, on the other hand, can be attributed to the presence of an α,β-unsaturated carbonyl group in its chemical structure, as observed by Marlou L.P.S. van Iersel et al. in 1998 [146]. Further studies have revealed that the inhibition of GSTP1-1 by ECA occurs through covalent binding of the GSH-ECA complex, leading to inhibitory activity in human tumor cells [128,144]. Interestingly, it has also been observed that ECA’s inhibition of GSTP1-1 in A549 and H460 lung cancer cells can increase the basal levels of p53. Additionally, ECA can enhance the anticancer effects of p53 by restoring its responsiveness via the inhibition of GSTP1, as noted by Yusuf and Srivenugopal in 2011 [61].

GSTO1 has been found to be highly expressed in esophageal squamous cell carcinoma (ESCC) and is believed to play an important role in tumor progression by regulating various cellular processes, including cancer cell proliferation, apoptosis, and migration [33,76,156]. Recent studies have also suggested that GSTO1 may be involved in the JAK/STAT3 signaling pathway, further highlighting its potential as a target for cancer therapy [33,156]. Notably, GSTO1-1 has been identified as one of the proteins overexpressed in cisplatin-resistant ovarian cancer cell lines, although direct evidence of its role in cancer cell drug resistance is lacking [157,158]. Interestingly, ECA has been shown to possess inhibitory activity against GSTO1 in cancer, albeit not as potent as the classical inhibitor S2E [33]. Given that GSTO1 is a crucial cellular antioxidant enzyme, the inhibition of its activity by ECA represents a promising avenue for tumor-targeted therapy [33].

GSH and GST are essential for safeguarding cells against electrophiles and chemicals. Through its catalytic binding to GSH, GST is known to protect tumor cells from the cytotoxic effects of certain drugs, as reported by Ruzza et al. (2009) [133]. Certain GSTs have been shown to detoxify alkylating agents such as melphalan through their catalytic binding to GSH [133]. Inhibition of GST activity by ECA can lead to changes in GSH content, which is known to increase in drug-resistant tumor cells [37,131,159,160,161,162] and decrease in non-resistant tumor cells [60]. In fact, ECA has been demonstrated to decrease GSH levels in colon cancer cells in a concentration-dependent manner, as noted by Ward et al. (2015) [60].

Tumor cells can develop resistance to anticancer drugs through various mechanisms, including changes in glutathione (GSH) and glutathione S-transferase (GST) activity levels. ECA has been found to increase GSH levels in the drug-resistant colon and non-small cell lung cancer cells, thereby resensitizing them to anticancer drugs [37,162]. ECA acts as a chemosensitizer targeting the GSH-dependent pathway and can enhance the cytotoxicity of other antitumor drugs, reversing chemoresistance and conferring therapeutic benefit to patients [131,133,136,154,163]. In vitro and in vivo studies have demonstrated that the combination of ECA with an alkylating agent can deplete cellular GSH levels and enhance the cytotoxicity of the alkylating agent in cancer cells, leading to a decrease in drug resistance [130,164].

The transmembrane transport of GSH conjugates is facilitated by the multidrug resistance protein MRP, which is known to be associated with drug resistance, as reported by Zaman et al. (1996) [145]. GSH, GST, and MRP1 have all been found to be involved in the pathway of anticancer drug resistance, according to O’Brien et al. (2000) [165]. MRP1 has been shown to be resistant to various natural drugs, as noted by Grant et al. (1994) [166].

ECA has been shown to impact MRP transcription and reverse MDR by inhibiting the expression of both MRP1 and MRP2, as noted by Ciaccio et al. (1996) [167]. In other words, ECA can react with GSH, catalyzed by GST, and the resulting GSH conjugate of ECA acts as an MRP substrate [165,167]. Zaman discovered that the ECA-GSH conjugate has the ability to inhibit MRP transport, which in turn impedes the release of drug conjugate from cells by MRP [145]. It is believed that this mechanism could help control drug resistance in tumor cells.

GSH serves a dual purpose as both a reducing agent and an antioxidant, as noted by Huang Lingping et al. (2017) [48]. ECA has been found to efficiently induce GSH depletion, thereby triggering a cascade of events. This process often leads to a concentration-dependent increase in intracellular ROS levels [52,134]. ECA has also been shown to effectively impact the Nrf2 signaling pathway, which regulates oxidative stress. GSH depletion induces the expression of the Nrf2-regulated gene HO-1 [52,168].

ECA was found to have an impact on apoptosis as a GST inhibitor. ECA-induced depletion of GSH resulted in mitochondrial dysfunction [52], which led to apoptosis in both NSC-34 motor neuron-like cells and lung cancer [48,52]. It has been noted by Franco that ECA’s depletion of GSH affects early apoptosis by modulating caspase activity through GSH efflux, mediated by an anion exchange mechanism [160,161].

### 4.4. Inhibition of the NF-κB Signaling Pathway

There is growing evidence that the NF-κB signaling pathway plays a crucial role in cancer development and progression by regulating processes such as cell proliferation, migration, and apoptosis [169]. Inflammatory diseases are often associated with increased expression of transglutaminase 2 (TGase 2), which can activate the NF-κB pathway [170].

ECA has been identified as an inhibitor of TGase-2 and the NF-κB pathway in a concentration-dependent manner [75,89]. By inhibiting TGase-2, ECA suppresses SPC-induced JNK activation and expression, inhibits SPC-induced K8 phosphorylation and recombination, and suppresses invasion and migration of PANC-1 pancreatic cancer cells [75,171].

The inhibition of TGase-2 by ECA may also interfere with the NF-κB signaling pathway, as evidenced by ECA’s ability to inhibit NF-κB-dependent signaling through various mechanisms, including the targeting of the cys38 thiol residue [89].

### 4.5. MAPK Signaling Pathway Inhibition

The mitogen-activated protein kinase (MAPK) pathway is a crucial signaling network that regulates diverse cellular processes, including cancer cell proliferation, differentiation, apoptosis, angiogenesis, and tumor metastasis [172]. Four eukaryotic MAPK cascades have been identified: ERK, JNK/stress-activated protein kinase, p38 MAPK, and ERK5 signaling pathways [172].

ECA has been identified as a modulator of the MAPK signaling pathway with different effects depending on its concentration [37]. The α,β-unsaturated ketone structure of ECA bears similarity to irreversible TKIs [30]. Studies using real-time PCR have demonstrated that ECA alone or in combination with TKI reduced the expression of MAPK-ERK1/2-mTOR downstream genes [30]. Additionally, previous studies have identified ECA as an inhibitor of the MAPK pathway [37,89].

In vitro and in vivo studies have shown that the combination of ECA and two EGFR TKIs at 25 μM inhibits breast cancer cell cycle progression and tumor growth by suppressing the MAPK-ERK1/2 signaling pathway [30]. ECA was also found to specifically inhibit lipopolysaccharide-induced ERK1/2 phosphorylation in RAW 264.7 macrophages [95]. However, some studies suggest that ECA treatment may increase ERK1 levels in colon cancer cells at concentrations of 25–75 μM [37]. ECA has also been identified as an inhibitor of the P38/MARK pathway through screening in a DNA-encoded small molecule library × protein library, with a significant reduction of p38 MAPK observed at a concentration of 100 μM [37,173].

Chan et al. discovered that ECA unexpectedly inhibits MAP2K6 kinase by targeting Cys38, although the function of Cys38 is currently unknown [173]. Dual phosphorylation of Thr180 and Tyr182 by MAP2K6 activates p38 MAPK signaling [174]. Thus, ECA may inhibit p38 MAPK signaling activation by targeting MAP2K6. Moreover, ECA has been found to inhibit JNK1 [37] and JNK activation and expression in PANC-1 cells through the inhibition of Tgase-2 [75].

### 4.6. Inhibition of STAT3 and HIF-1 Signaling Pathways

HIF-1 and STAT3 are recognized as key inducers of VEGF expression and are upregulated in multiple cancer types [175,176]. STAT3 also plays a critical role in regulating immune evasion, making it an attractive target for cancer therapy [176]. Small molecule inhibitors targeting STAT3 have been shown to suppress HIF-1 and VEGF expression, as well as impede tumor growth and angiogenesis in vivo [175].

Recently, ECA has been identified as a potent STAT3 inhibitor using the CMap method [32]. ECA effectively reduced STAT3 phosphorylation in a dose- and time-dependent manner, although it did not inhibit STAT3 upstream kinases. In prostate cancer DU145 cells, ECA activates protein tyrosine phosphatase 1B (PTP1B) and SH2 domain-containing protein tyrosine phosphatase 2 (SHP2), leading to the inhibition of STAT3 activity and downregulation of the STAT3-responsive gene cyclin D1 [32]. It was confirmed that ECA activates SHP2 through phosphorylation at the Y580 site and directly binds to SHP2, but the mechanism by which ECA activates PTP1B remains unknown [32].

Additionally, ECA has been found to inhibit the interaction between p300 and HIF-1a, blocking the HIF signaling pathway and reducing HIF-1a transcriptional activity in HeLa cells under hypoxic conditions [83]. These findings suggest that ECA may hold therapeutic potential in cancer treatment by targeting multiple signaling pathways involved in tumor angiogenesis and metastasis.

### 4.7. NOTCH Signaling Pathway Inhibition and Induction of Oxidative Stress Pathway

In human cancers, various components of the Notch signaling cascade are upregulated or activated [177]. Notably, Myc has been shown to contribute to the Notch-induced tumorigenesis process, and it is a direct target of the Notch pathway [178]. Although Notch1 expression has not been directly observed, ECA has been found to inhibit tumor growth in human non-muscle-invasive bladder cancer (NMIBC) cell lines by inhibiting cell proliferation and inducing apoptosis through the NOTCH signaling pathway [179]. Moreover, in chronic lymphocytic leukemia (CLL) cells and multiple myeloma MPC11 cells, ECA suppressed Myc expression [38,43,92]. These findings suggest that ECA may have therapeutic potential in the treatment of cancer by targeting the Notch signaling cascade and Myc.

Redox homeostasis is frequently disrupted in human diseases, particularly in cancer. The Keap1-Nrf2-ARE pathway, which is activated by oxidative stress, is a major signaling pathway [180]. The transcription factor Nrf2 regulates the expression of antioxidant and detoxification genes, such as HO-1 [181]. Imbalanced levels of HO-1 have been implicated in the pathogenesis of cancer [182]. Nrf2 small molecule compounds that activate the Nrf2 signaling pathway could be used as cancer chemopreventive drugs [183].

ECA is a redox-active, chemoprotective, and cytotoxic drug with an active electrophilic center [168,184]. ECA activates the Nrf2 signaling pathway [168,184]. There is evidence that ECA increases the expression of the HO-1 protein in PBMCs, and its direct interaction with Keap1 activates the Nrf2 pathway in CLL cells [184]. ECA also triggers cellular DNA damage by inducing oxidative stress [60]. In colon cancer HCT116 cells, ECA resulted in increased nuclear expression of Nrf-2 and increased mRNA expression of the redox-sensitive stress genes HO-1, GADD153, and p21 [60].

## 5. A New Analog of Ethacrynic Acid

ECA can introduce additional hydrophobic side chains through its carboxylic acid moiety, enhancing ECA’s own poor antiproliferative ability [185,186]. Given the limitations of ECA administration, developing ECA derivatives is an appropriate approach. Various ECA derivatives with different activities have been synthesized, some of which have been marketed as anticancer drugs [187,188].

The α,β-unsaturated carbonyl groups of ECA mainly inhibit the activity of GST P1-1, while the carboxyl group mainly modulates its diuretic side effects [187,189]. The α,β-unsaturated carbonyl groups of ECA are important active groups in ECA derivatives (Figure 5) [186,187,190,191]. Replacement of chlorine at the ortho-position of α, β-unsaturated carbonyl with hydroxyl promotes the formation of intramolecular hydrogen bonds and enhances the Michael acceptor chemical reactivity of ECA derivatives, thereby conferring antiproliferative activity of the newly synthesized ECA analogs on cancer cells [191]. 

Carboxyl modification: The carboxylic acid part of ECA can be substituted with a heterocycle, ester, amide, aromatic ester, or aromatic amide, oxadiazole, triazole, sulfonamide, thiazole, glucosamine, etc., to achieve high antiproliferative activity and cancer cell selectivity (Figure 5) [17,128,186,189,192,193,194,195,196,197,198,199]. 

Ester derivatives have lower antiproliferative activity than amide derivatives [186]. Piperazine or 4-aminopiperidine is the best linker for carboxyl groups [194]. Oxadiazole ECA derivatives have the highest antiproliferative ability [186,195,196,199]. Oxadiazole derivatives exhibit potent antitumor activity in vitro and in vivo, inhibit the proliferation of human SW620 colon cancer in xenograft in vivo, and are promising antitumor drugs [189]. Experimental results of 6r (5-[2,3-Dichloro-4-(2-methylene-1-oxobutyl) phenoxymethyl]-3-methyl-1,2,4-oxadiazole) suggest that the antiproliferative effects of ECA oxadiazole analogs are related to cell cycle arrest [189].

ECA-triazole derivatives, ECA-oxadiazole derivatives, ECA-amide derivatives, ECA-thiazole derivatives, and ECA-glucosamine derivatives all enhance their ability to inhibit GSTpi activity [128,186,194,195,197,199], leading to caspase-induced apoptosis. Compared with ECA, ECA glucosamine derivatives significantly reduce diuretic activity, have better cancer cell selectivity, block the G2/M cell cycle, and have strong antiproliferative effects in cancer cells, which may be due to the amino group of EGA. It is caused by glucose binding [128,197].

Modification of unsaturated double bond units and phenyl rings: the introduction of an alkyl chain in place of a double bond replaces the two chlorine or hydrogen atoms in the benzene ring with another substituent (Figure 5) [185]. Preliminary studies have indicated that certain analogs of ethacrynic acid (ECA), which lack α,β-unsaturated carbonyl groups, can inhibit human MCF-7/AZ breast cancer cell migration without exhibiting diuretic effects [200]. The ECA derivative para-acylated m-cresol, synthesized by Zack E. Bryant et al., was found to significantly inhibit migration by up to 65% in human cancer cell lines C4–2B and Hs578Ts(i)8 [77].

However, these analogs lacking α,β-unsaturated carbonyl units do not exhibit any antiproliferative activity [77,78]. ECA analogs with different substituents on the benzene ring exhibit varying rates of cancer cell migration and can inhibit breast cancer cell migration by up to 52%, making them an excellent model system for studying migration and invasion [77,78]. Modified ECA analogs with unsaturated double bond units have the potential to act as novel GST inhibitors and exhibit better inhibition than ECA [199,200,201]. Additionally, substituents such as chlorine, bromine, fluoride, or methyl on the phenyl group have been shown to inhibit GST P1-1 activity [199,200,202].

ECA derivatives of transition metal complexes, such as platinum (Pt), ruthenium (Ru), and osmium (Os) complexes (Figure 6), have been shown to enhance the anticancer properties of the resulting complexes and inhibit GST activity. Ruthenium complexes, in particular, have demonstrated superior anticancer properties, with increased selectivity and reduced toxicity towards cancer cells, compared to conventional platinum-based drugs. EthaRAPTA, a potential breast cancer treatment, has high selectivity, reduced general toxicity, multiple mechanisms to induce apoptosis, and has been shown to overcome resistance to platinum-based therapies. While the exact mechanism of action is unclear, EthaRAPTA has demonstrated enhanced antiproliferative activity.

Osmium (Os) complexes are potent growth inhibitors of human cancer cells with significant selectivity and are considered interesting alternatives to ruthenium-based anticancer agents. Ruthenium and osmium complexes have been found to be consistent with previous mechanistic studies of platinum cancer metal drugs. While conjugation of ECA with transition metals has great potential as an anticancer drug, not all transition metal conjugates can enhance the antitumor effect of ECA itself. Therefore, further studies on drug design are needed.

Applications of nanoformulations: the functional group of ECA has developed new nanodevices that can inhibit solid and liquid tumor growth by improving the physicochemical properties (PK/PD behavior) of ECA, enhancing its antiproliferative activity, and increasing its ability to inhibit GST activity. These nanodevices were covalently grafted onto the surface of the phosphorus dendrimer [185,186].

To overcome drug resistance in platinum-based chemotherapy, Qiang Yang et al. proposed the construction of biodegradable nanoparticles and found that the resulting mixed nanomedicine effectively inhibited the growth of breast cancer in mice with better safety [41]. Li et al. developed M-ECA-Pt, a nanopharmaceutical formulation that encapsulates etacraplatin into nanoparticles for cancer treatment [128,203].

The administration of mixed nanoparticles can sufficiently improve the drug’s original water solubility, enhance anticancer efficacy, reduce toxicity, and effectively improve cisplatin resistance [47,128,203]. Moreover, a non-diuretic, brain-permeable, cancer cell-selective, hydrophobic ECA derivative, KSS72, has demonstrated antitumor effects in an orthotopic glioblastoma (GBM) model and has the potential to enter clinical trials [204]. ECA derivatives have also been reported to act as covalent bromodomain inhibitors, regulating cancer cell growth and apoptosis [205]. The synergistic anticancer performance of clinical photodynamic therapy (PDT) under hypoxic conditions can be enhanced when ECA is covalently linked to brominated body photosensitizers (BPS) to form conjugated ECA-BPS [10].

## 6. Perspectives

ECA was reported as an antitumor drug after researchers repurposed the traditional circulatory diuretic. We summarized the effects of ECA on several cancer traits and explored the mechanisms underlying its effects. ECA has exhibited diverse antitumor activities in various in vitro cancer cells and in vivo mouse tumor models. Specifically, at concentrations that did not affect cancer cell growth, ECA also inhibited EMT processes, migration and invasion processes, and angiogenesis in cancer and resisted the development of inflammation. However, there is no direct evidence of a relationship between ECA and certain cancer markers.

We speculate that ECA may affect the tumor microenvironment, energy metabolism, immune evasion, replicative immortality, and genomic instability in cancer cells. The antitumor effects of ECA in vitro and in vivo are regulated by various mechanisms, including Ca^2+^-gated channels, inhibition of GST activity, and the Wnt signaling pathway.

In addition, combining ECA with an alkylating agent has the potential to be effective in antitumor therapy. Combination therapy has emerged as an increasingly useful cancer treatment strategy. By choosing an appropriate combination of drugs that can create a synergistic effect, the antitumor effects of combination drugs are often better than single drugs [41]. ECA has also been identified as a potential candidate for the reversal of chemotherapy resistance and enhancement of the cytotoxicity of other antitumor drugs.

However, the activity of ECA is limited by its carboxylic acid moiety, which impedes its ability to penetrate cell membranes. To overcome this limitation, a number of cancer cell-selective ECA derivatives have been synthesized, which show enhanced bioavailability, inhibit the activity of GST, and exhibit antiproliferative and antimigration effects on cancer cells. Moreover, some ECA derivatives have also demonstrated the ability to inhibit tumor growth in vivo. While synthesized ECA and transition metal conjugated derivatives exhibit promising anticancer properties, not all transition metal conjugates enhance the antitumor effect of ECA itself. Therefore, there is a need for further drug design studies.

In summary, ECA and its derivatives offer a safer and more effective approach to cancer treatment, with the potential to exert anticancer effects.

## Figures and Tables

**Figure 1 ijms-24-06712-f001:**
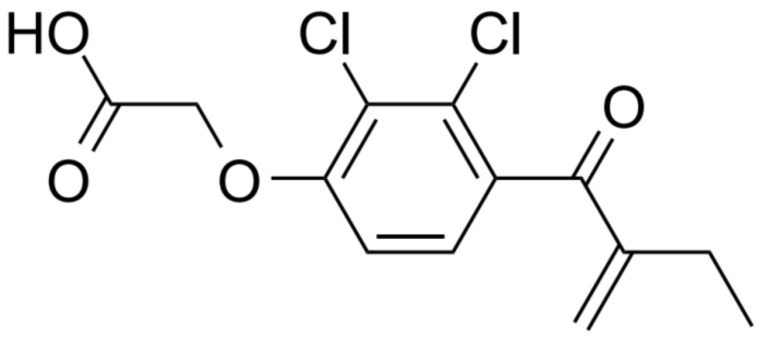
Chemical structures of the ethacrynic acid (ECA).

**Figure 2 ijms-24-06712-f002:**
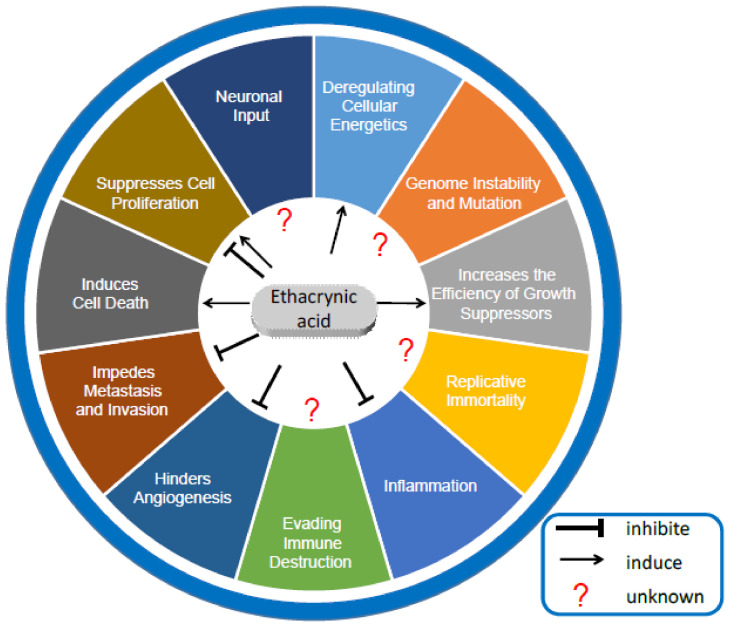
Effects of ECA on hallmarks of cancer.

**Figure 3 ijms-24-06712-f003:**
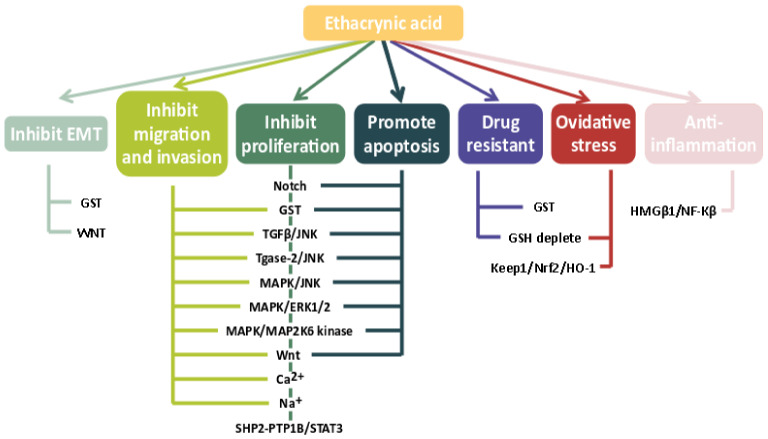
Effect and mechanism of ECA on cancer cells.

**Figure 4 ijms-24-06712-f004:**
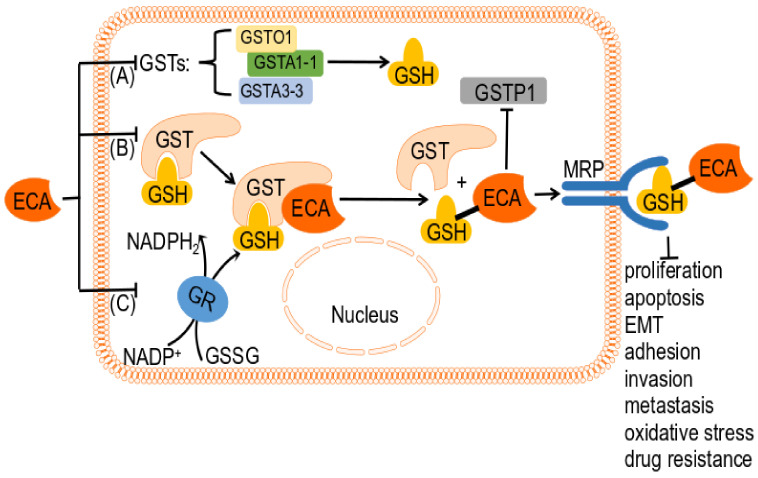
Possible GSH mechanism of ECA in cancers. (A) ECA directly inhibits the activity of GSTs, such as GSTO1, GSTA1-1, and GSTA3-3. (B) ECA can be catalyzed by GST, covalently bound to GSH, and the resulting GSH-ECA conjugate acts as an MRP substrate and inhibits MRP transport. The GSH-ECA complex also inhibits GSTP1-1, resulting in suppressive activity in human tumor cells. (C) ECA affects glutathione reductase (GR) by modulating intracellular GSH levels. Abbreviations: ECA, ethacrynic acid; GST, glutathione S-Transferase; GSH, glutathione; GSTO1, glutathione S-transferase omega-1 (GSTO1); GSTA1-1, glutathione S-transferase alpha 1-1; GSTA3-3, glutathione S-transferase alpha 3-3; GSTP1, glutathione S-transferase Pi 1; GR, glutathione reductase; GSSG, oxidized glutathione; MRP, multidrug resistance protein.

**Figure 5 ijms-24-06712-f005:**
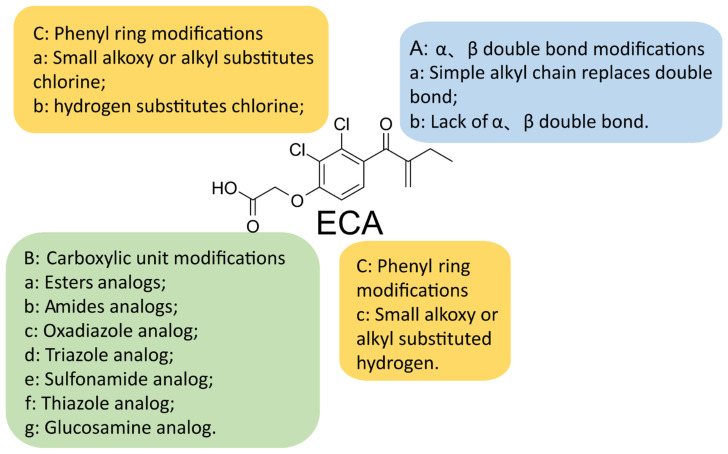
The site of chemical modification of ethacrynic acid derivatives.

**Figure 6 ijms-24-06712-f006:**
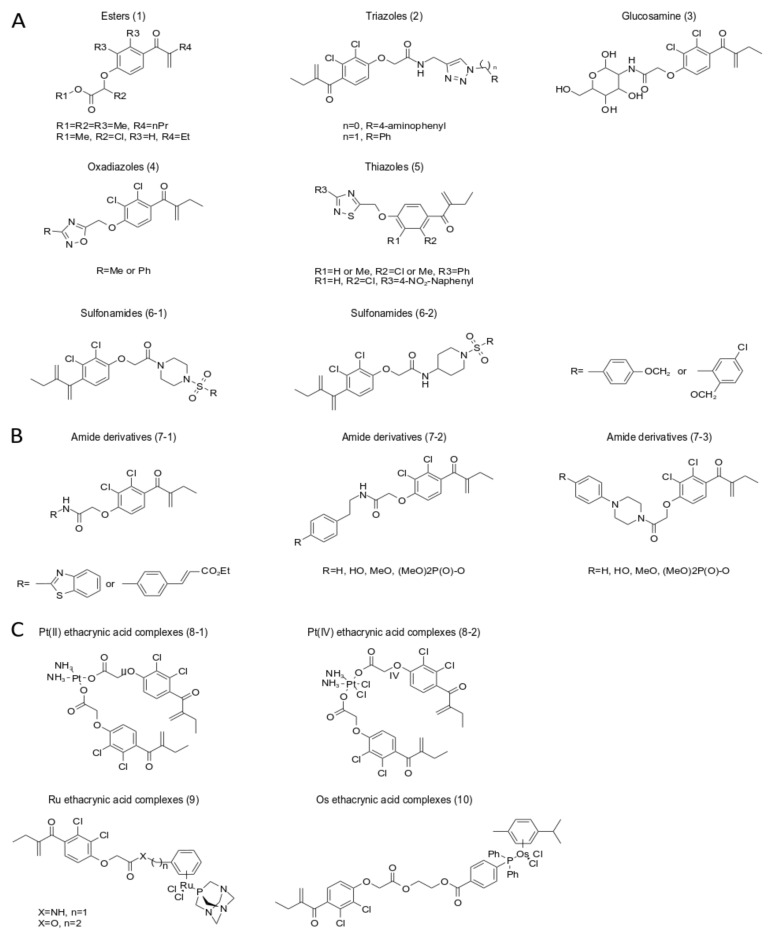
Chemical structures of several major ECA derivatives with anticancer potential. (**A**) Structure of ECA derivatives containing ester, triazole, glucosamine, oxadiazole, thiazole, sulfonamide. (**B**) Structure of ECA derivatives containing amide. (**C**) Structure of ECA derivatives of transition metal complexes, such as platinum (Pt), ruthenium (Ru), and osmium (Os) complexes.

**Table 1 ijms-24-06712-t001:** Half-maximal inhibitory concentration (IC_50_) of ECA in different human and mouse tumor cell lines.

Types of Cancer	Cell Lines	The IC_50_ of ECA (µmol/L)/h *	References
Lung cancer	A549	87.03/48 h	[38]
	A549	178/48 h	[39]
	H1975	99.54/48 h	[38]
Pancreatic cancer	DanG	67.8/not indicated	[31]
	PancO2 (mouse)	141.7/not indicated	[31]
Malignant Melanoma	A375	57.26 ± 6.6/not indicated	[40]
	SK-Mel-28	122/48 h	[39]
	B16 (mouse)	201/48 h	[39]
cervical cancer	HeLa	127.2 ± 13.1/not indicated	[40]
breast cancer	MCF7	45.53/24 h	[30]
	MCF7	63/48 h	[39]
	MCF7	340/48 h, 475/72 h	[41]
	MDA-MB-231	39.64/24 h	[30]
	MDA-MB-231	42.74 ± 0.93/48 h	[33]
	MDA-MB-468	39.04 ± 1.12/48 h	[33]
	4T1 (mouse)	25.23/24 h	[30]
Prostate Cancer	LNCap	46/48 h	[39]
	PC3	67/48 h	[39]
Colon Cancer	HCT116	58/48 h	[39]
	SW480	68/48 h	[39]
	HT29	56/48 h	[39]
Hepatocellular carcinoma	HepG2	223/48 h	[39]
	HepG2	14.8/not indicated	[42]
	Hep3B	6.4/not indicated	[42]
Multiple myeloma	OPM-2	22/72 h	[36,43]
	U266	90/48 h	[39]
	U266	60/72 h	[36,43]
	RPMI-8226	8/72 h	[36,43]
	KMS-18	7/not indicated	[36]
Lymphoma	Raji	33/not indicated	[36]
	OCI-Ly8 LAM53	57/not indicated	[36]
	SU-DHL 4	58/not indicated	[36]
	RAMOS	174/48 h	[39]
Plasmocytoma	MPC-11 (mouse)	50/72 h	[36,43]
Kidney cancer	A498	50/72 h	[44]
	A704	150/72 h	[44]
	Caki-2	70/72 h	[44]

h* represents the treatment time in hours (h).

**Table 2 ijms-24-06712-t002:** The classification of GSTs in human.

GSTs Classification	Members
alpha (A)	GSTA1-GSTA4
kappa (K)	GSTK1
mu (M)	GSTM1-GSTM5
omega (O)	GSTO1
pi (P)	GSTP1
sigma (S)	GSTS1
theta (T)	GSTT1, GSTT2
zeta (Z)	GSTZ1

## Data Availability

Not applicable.

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
