# Peer review of "Ethacrynic Acid: A Promising Candidate for Drug Repurposing as an Anticancer Agent"

_ijms, 2023, doi:10.3390/ijms24076712_

Round 1

Reviewer 1 Report

In this manuscript, Lo Yu and coworkers wrote a review about Ethacrynic acid, a well-known diuretic, and its possible repurposing as an anticancer agent. The authors have made thorough literature survey and this review is potentially useful to the scientific community working in the field. The manuscript is well organized, and literature is up to date.

However, I have noticed some discrepancies with literature data. The authors should check data presented in the first paragraph of Introduction with data in references 1-3. On page 6, line 162 authors wrote “while it had the greatest effect on the apoptosis of colon cancer cells at  a concentration of 200 μM”, I was not able to find this data in quoted paper (reference 37).

Before publishing some editorial work is needed. For example, throughout all text charges on ions must be written as superscripts. Also, chemical formulae of ECA (page 2) should be correctly written. Caption on Figure one is needed.

Author Response

Dear Guest Editor  

We are grateful for your consideration of our manuscript entitled “Ethacrynic Acid: A Promising Candidate for Drug Repurposing as an Anticancer Agent” (manuscript ID: ijms-2276553) by Yu et al. and appreciate your helpful comments.

Thank you for your comment. We have revised the text and add figure according to your comments. Red characters were used to highlight improved parts of the revised manuscript.

Replies to the comments are as follows:

[To reviewer 1]

In this manuscript, Lo Yu and coworkers wrote a review about Ethacrynic acid, a well-known diuretic, and its possible repurposing as an anticancer agent. The authors have made thorough literature survey and this review is potentially useful to the scientific community working in the field. The manuscript is well organized, and literature is up to date.

However, I have noticed some discrepancies with literature data.

1) The authors should check data presented in the first paragraph of Introduction with data in references 1-3.

Response) Thank you for comments. We mistakenly used the number of cancer cases in 2020-2022 years in references 1-3. However now the latest cancer statistics 2023 have been published. So we directly use the latest data in 2023 instead of it.

We delete that “The latest data estimates that in the United States alone, 1,918,030 new cancer cases and 608,570 cancer-related deaths will occur by 2021 [1]. The figures for 2022 indicate a further 20,000 cases and 110,000 deaths compared to 2021, highlighting the urgent need for effective treatment options [1-3].”

[1] Siegel, R.L., et al., Cancer statistics, 2022. CA: a cancer journal for clinicians, 2022. 72(1): p. 7-33.

[2] Siegel, R.L., et al., Cancer Statistics, 2021. CA Cancer J Clin, 2021. 71(1): p. 7-33.

[3] Siegel, R.L., K.D. Miller, and A. Jemal, Cancer statistics, 2020. CA Cancer J Clin, 2020. 70(1): p. 7-30.

We have changed it to “The latest data estimates that in the United States alone, 1,958,310 new cancer cases and 609,820 cancer-related deaths will occur by 2023 [1]. The figures for 2023 indicate a further 40,280 new cases compared to 2022, highlighting the urgent need for effective treatment options [1, 2].”

[1]. Siegel, R.L.; Miller, K.D.; Wagle, N.S.; Jemal, A. Cancer statistics, 2023. CA Cancer J. Clin. 2023, 73, 17-48.

[2]. Siegel, R.L.; Miller, K.D.; Fuchs, H.E.; Jemal, A. Cancer statistics, 2022. CA Cancer J. Clin. 2022, 72, 7-33.

2) On page 6, line 162 authors wrote “while it had the greatest effect on the apoptosis of colon cancer cells at a concentration of 200 μM”, I was not able to find this data in quoted paper (reference 37).

Response) Thank you for excellent comments. Sorry for mistakes. Actually, this data comes from Ethacrynic Acid and Ciclopiroxolamine in Various Cancer Cells, which the reference 26.

We have changed it to “ECA induced cell death in human colon cancer DLD-1 cells at high concen-trations of 60-100 µM [37]), while it had the greatest effect on the apoptosis of colon cancer cells at a concentration of 200 µM [27].”

[Figure attached]

3) Before publishing some editorial work is needed. For example, throughout all text charges on ions must be written as superscripts.

Response) Thank you for excellent comments. We have changed all ions charges as superscript writing in all texts as shown below.

(1) We have changed “Na+” to “and increases the transport of Na+ into the distal tube” on line 67.

(2) We have changed “Ca2+ homeostasis is associated with telomerase activity” to “ evidence suggests that Ca2+ homeostasis is associated with telomerase activity” on line 203.

(3) We have changed “This inhibition of Na+, K+ activation, Mg2+-dependent ATPase activity by ECA is irreversible.” to “This inhibition of Na+, K+ activation, Mg2+-dependent ATPase activity by ECA is irreversible.” on line 261, 262.

(4) We have changed “Na+/K+/2Cl- cotransporter” to “Na+/K+/2Cl- cotransporter” on Line 431, 443.

(5) We have changed “Ca2+-gated channels” to “including Ca2+-gated channels, inhibition of GST activity, and the Wnt signaling pathway.” on line 849.

4) Also, chemical formulae of ECA (page 2) should be correctly written.

Response) Thank you for excellent comments. We have changed “C13H12Cl2O4” to “C13H12Cl2O4 ” on line 58 in page 2

5) Caption on Figure one is needed.

Response) Thank you for excellent comments. We have changed to “Figure 1. The chemical structure of ethacrynic acid.” on line 61.

Reviewer 2 Report

The submitted manuscript "Ethacrynic Acid: A Promising Candidate for Drug Repurposing as an Anticancer Agent” is research focused on the biological application of ethacrynic acid specially as anticancer agent. This work is a valuable perfect, and suitable contribution to be published in International Journal of Molecular Sciences after justifying some points 

Form and content comments 

In vitro and in vivo should be written in italics in all the manuscript 

IC50 should be written instead IC50 in all the manuscript 

Homogenize the antiproliferative in all the manuscript 

Line 58 : C13H12Cl2O4 should be written instead C13H12Cl2O4  

Line 67, 203, 431, 443 and 848 : Na+ should be written instead Na+ 

Table 1 : IC50 of ECA(μmol/L)/h  should be written instead IC50 of ECA(μmol/L 

Line 747 and 749 : α,β-unsaturated should be written instead α and β-unsaturated 

Line 749 and 750  : Regarding to the important of α,β-unsaturated group can you improve this section and collected data with reference (https://doi.org/10.3390/molecules28020910

Line 750 and 760 : Figure 5 should be written instead Figure 4 

Line 767: maybe the structure of 6r should be given 

Line 779 : Figure 5 should be written instead Figure 4 

The paragraph lines 780-782 should be revised with reference 186 

Line 795: Figure 6 should be written instead Fig 5 

Homogenize antitumor in all the manuscript 

Structures in figure 6 are to small, the size of the atoms should be increased 

The abbreviation of every journal in the references part should be written according to the journal style   

According to the above comments, the article needs major revision. So I accept the article to be publishing in International Journal of Molecular Sciences after correction 

 Best wishes  

Author Response

Dear Guest Editor  

We are grateful for your consideration of our manuscript entitled “Ethacrynic Acid: A Promising Candidate for Drug Repurposing as an Anticancer Agent” (manuscript ID: ijms-2276553) by Yu et al. and appreciate your helpful comments.

Thank you for your comment. We have revised the text and add figure according to your comments. Red characters were used to highlight improved parts of the revised manuscript.

Replies to the comments are as follows:

[To reviewer 2]

The submitted manuscript "Ethacrynic Acid: A Promising Candidate for Drug Repurposing as an Anticancer Agent” is research focused on the biological application of ethacrynic acid specially as anticancer agent. This work is a valuable perfect, and suitable contribution to be published in International Journal of Molecular Sciences after justifying some points 

Form and content comments 

  1. In vitro and in vivo should be written in italicsinall the manuscript 

 - Response: Thank you for excellent comments. We have rewritten them in italics in all the manuscript as shown below.

  • Line 117: ECA has been demonstrated to possess a significant in vivo inhibitory effect on tumor growth in several studies.
  • Line 468: In a study of EGFR mutant lung cancer, Nakayama and colleagues (2014) found that ethacrynic acid can reduce lung tumor formation in an in vivo mouse model by inhibiting β-catenin in the Wnt signaling pathway.
  • Line 520: Despite the promising results of in vitro experiments using ECA to suppress GST regulation and improve response to anticancer drugs, clinical success has been hindered by the side effects of ECA, and further development of GST inhibitors is warranted [137, 138].
  • Line 528: In vitro and in vivo studies have demonstrated that ECA can reverse drug resistance phenotypes.
  • Line 615: In vitro and in vivo studies have demonstrated that the combination of ECA with an alkylating agent can deplete cellular GSH levels and enhance the cytotoxicity of the alkylating agent in cancer cells, leading to a decrease in drug resistance [129, 163].
  • Line 673: In vitro and in vivo studies have shown that the combination of ECA and two EGFR TKIs at 25 μM inhibits breast cancer cell cycle progression and tumor growth by suppressing the MAPK-ERK1/2 signaling pathway [29].
  • Line 696: Small molecule inhibitors targeting STAT3 have been shown to suppress HIF-1 and VEGF expression, as well as impede tumor growth and angiogenesis in vivo [174].
  • Line 762-764: Oxadiazole derivatives exhibit potent antitumor activity in vitro and in vivo, inhibit the proliferation of human SW620 colon cancer in xenograft in vivo, and are promising antitumor drugs [188].
  • Line 838: ECA has exhibited diverse antitumor activities in various in vitro cancer cells and in vivo mouse tumor models.
  • Line 846: The antitumor effects of ECA in vitro and in vivo are regulated by various mechanisms, including Ca2+-gated channels, inhibition of GST activity, and the Wnt signaling pathway.
  • Line 863: Moreover, some ECA derivatives have also demonstrated the ability to inhibit tumor growth in vivo.
  1. IC50 should be written instead IC50in all the manuscript 

Response: Thank you for excellent comments. We have changed “IC50” to “IC50 ” on line 102, 110, and Table 1 as shown below.

  • We have changed “IC50 values range from 6-223 μM, Table 1” to “IC50 values range from 6-223 μM, Table 1 ” on line 102.
  • We have changed “Table 1. Half-maximal inhibitory concentration (IC50) of ECA in different human and mouse tumor cell lines.” to “Table 1. Half-maximal inhibitory concentration (IC50) of ECA in different human and mouse tumor cell lines.” on line 110.
  • We have changed “The IC50 of ECA(µmol/L)” to “The IC50 of ECA(µmol/L)/h*” on Table 1. And we add “h* represents the treatment time in hours (h)”.
  1. Homogenize the antiproliferative in all the manuscript 

Response: Thank you for excellent comments. We have corrected all “anti-proliferative” to “antiproliferative”in the manuscript on line 104, 107, 133 and 861. The antiproliferative was homogenized in all the manuscript.

  1. Line 58 : C13H12Cl2O4 should be written instead C13H12Cl2O4

Response: Thank you for excellent comments. We have changed “C13H12Cl2O4” to “ C13H12Cl2O4 ” on line 58.

  1. Line 67, 203, 431, 443 and 848: Na+ should be written instead Na+ 

 - Response: Thank you for excellent comments. We have changed all ions charges as superscript writing in all texts as shown below.

  • We have changed “Na+” to “and increases the transport of Na+ into the distal tube” on line 67.
  • We have changed “Ca2+ homeostasis is associated with telomerase activity” to “ evidence suggests that Ca2+ homeostasis is associated with telomerase activity” on line 203.
  • We have changed “This inhibition of Na+, K+ activation, Mg2+-dependent ATPase activity by ECA is irreversible.” to “This inhibition of Na+, K+ activation, Mg2+-dependent ATPase activity by ECA is irreversible.” on line 261, 262.
  • We have changed “Na+/K+/2Cl- cotransporter” to “Na+/K+/2Cl- cotransporter” on Line 430, 442.
  • We have changed “Ca2+-gated channels” to “including Ca2+-gated channels, inhibition of GST activity, and the Wnt signaling pathway.” on line 847.
  1. Table 1 : IC50 of ECA(μmol/L)/h  should be written instead IC50 of ECA(μmol/L

 - Response: Thank you for excellent comments. We changed to “IC50 of ECA(μmol/L)/h*”.  We add “h* represents the treatment time in hours (h)”.

  1. Line 747 and 749 : α,β-unsaturated should be written instead α and β-unsaturated

Response) Thank you for excellent comments. We found that line 747 and 749 are written “The α and β-unsaturated”. You mean maybe replace “The α and β-unsaturated” with “The α,β-unsaturated” on line 747 and 749? We have changed “The α and β-unsaturated” to “The α,β-unsaturated” on line 747 and 749.

  1. Line 749 and 750  : Regarding to the important of α,β-unsaturated group can you improve this section and collected data with reference (https://doi.org/10.3390/molecules28020910) 

Response) Thank you for excellent comments. First of all, we added this reference to the original 749-750 line sentence. And we add the discussion on line 748-753 as below to make the content more clear.

“The α,β unsaturated carbonyl groups of ECA are important active groups in ECA derivatives (Figure 5) [183,185,188,189]. Replacement of chlorine at the ortho-position of α, β-unsaturated carbonyl with hydroxyl promotes the formation of intramolecular hydrogen bonds and enhances the Michael acceptor chemical reactivity of ECA derivatives, thereby conferring antiproliferative activity of the newly synthesized ECA analogues on cancer cells [189].”

[189] El Abbouchi, A.; El Brahmi, N.; Hiebel, M.-A.; Ghammaz, H.; El Fahime, E.; Bignon, J.; Guillaumet, G.; Suzenet, F.; El Kazzouli, S. Improvement of the Chemical Reactivity of Michael Acceptor of Ethacrynic Acid Correlates with Antiproliferative Activities. Molecules 2023, 28, 910.

  1. Line 750 and 760: Figure 5 should be written instead Figure 4 

Response) Thank you for excellent comments. We have changed “Figure 5” to “Figure 4” on line 750 and 760.

  1. Line 767: maybe the structure of 6r should be given

Response) Thank you for excellent comments. Considering that it is not very beautiful to make a picture of 6r alone, and 6r is only mentioned once in the full text. So we added the chemical structure of 6r to the sentence on line 766-769, when it was first mentioned in the manuscript, as shown below.

“Experimental results of 6r (5-[2,3-Dichloro-4-(2-methylene-1-oxobutyl) phe-noxymethyl]-3-methyl-1,2,4- oxadiazole) suggest that the antiproliferative ef-fects of ECA oxadiazole analogs are related to cell cycle arrest [189].”

  1. Line 779: Figure 5 should be written instead Figure 4

Response) Thank you for excellent comments. We have changed “Figure 4” to “Figure 5” on line 779.

  1. The paragraph lines 780-782 should be revised with reference 186

Response) Thank you for excellent comments. We checked this sentence on lines 780-782 and found that it was a misused reference, which is [186]. So instead of revising the sentence, I have corrected it to the correct reference [76], as follows.

“Preliminary studies have indicated that certain analogs of ethacrynic acid (ECA), which lack α,β-unsaturated carbonyl groups, can inhibit human MCF-7/AZ breast cancer cell migration without exhibiting diuretic effects [200].”

Old reference: [186] Mignani, S.; El Brahmi, N.; El Kazzouli, S.; Eloy, L.; Courilleau, D.; Caron, J.; Bousmina, M.M.; Caminade, A.-M.; Cresteil, T.; Majoral, J.-P. A novel class of ethacrynic acid derivatives as promising drug-like potent generation of anticancer agents with established mechanism of action. European Journal of Medicinal Chemistry 2016, 122, 656-673.

New reference: [200] Janser, R.F.; Meka, R.K.; Bryant, Z.E.; Adogla, E.A.; Vogel, E.K.; Wharton, J.L.; Tilley, C.M.; Kaminski, C.N.; Ferrey, S.L.; Steelant, W.F. Ethacrynic acid analogues lacking the α, β-unsaturated carbonyl unit—Potential anti-metastatic drugs. Bioorg. Med. Chem. Lett. 2010, 20, 1848-1850.

  1. Line 795: Figure 6 should be written instead Fig 5 

Response) Thank you for excellent comments. We have changed “Fig. 5” to “Figure 6” on line 794.

  1. Homogenize antitumor in all the manuscript 

Response) Thank you for excellent comments. We have corrected all “anti-tumor” to “antitumor”in the manuscript on line 43, 827, 852, 857, and 861. The antitumor was homogenized in all the manuscript.

  1. Structures in figure 6 are to small, the size of the atoms should be increased

Response) Thank you for excellent comments.  We have modified the structure in figure 6 to the standard ACS document 1996 format, thereby increasing the size of the atoms, as shown below.

Old

[Figure attached]

Figure 6. Chemical structures of several major ECA derivatives with anticancer potential.

New:

[Figure attached]

Figure 6. Chemical structures of several major ECA derivatives (1-10) with anticancer potential.

  1. The abbreviation of every journal in the references part should be written according to the journal style

Response) Thank you for excellent comments. We have changed it.   

  1. Some bugs corrected, such as:

“ECA can introduce additional hydrophobic side chains through its carboxylic acid moiety, enhancing ECA's own poor antiproliferative ability [183,184].” This sentence is repeated twice on line 741-742 and line 752-754, we deleted once on line 752-754.
